# SSFX (Space Sound Effects) Short Film Festival: Using the film festival model to inspire creative art-science and reach new audiences

Martin O. Archer[1]

[1]School of Physics and Astronomy, Queen Mary University of London, London, UK

**Correspondence:** Martin O. Archer
(m.archer@qmul.ac.uk)

**Abstract.** The ultralow frequency analogues of sound waves in Earth's magnetosphere play a crucial role in space weather, however, the public is largely unaware of this risk to our everyday lives and technology. As a way of potentially reaching new audiences, SSFX made 8 years of satellite wave recordings audible to the human ear with the aim of using it to create art. Partnerting with film industry professionals, the standard processes of international film festivals were adopted by the project in order to challenge independent filmmakers to incorporate these sounds into short films in creative ways. Seven films covering a wide array of topics/genres (despite coming from the same sounds) were selected for screening at a special film festival out of 22 submissions. The works have subsequently been shown at numerous established film festivals and screenings internationally. These events have attracted diverse non-science audiences resulting in several unanticipated impacts upon them, thereby demonstrating how working with the art world can open up dialogues with both artists and audiences who would not ordinarily engage with science.

## 1 Introduction

Public engagement projects which see artists and scientists collaborate together in some way have become increasingly popular, with growing evidence that such projects, through a variety of methods, contribute to the development of society (Malina et al., 2018). Malina (2010) classifies such collaborations into the following categories:

I. Scientists collaborating with artists on common projects resulting in both scientific discoveries and the production of art works

II. Scientists applying scientific research to understand creative activity in the arts

III. Scientists working with artists to develop technological inventions

IV. Working as both a scientist and an artist in a dual career

V. Scientists engaging with the arts to enhance cultural appropriation of science

## VI. Scientists engaging with the arts to improve the ways science is communicated to the public

To better understand scientists' motivations for such endeavours at an art-science session at the 2019 Interact symposium (Archer et al., 2019) 12 university science researchers and public engagement professionals were surveyed for their attitudes towards art-science collaborations. This found that their interest in art-science collaborations were based on enjoyment of both subjects, utilising their creativity, and as a communication tool (particularly for different audiences). When asked in an open question who they thought the audience was for such collaborations: $67 \pm 17\%$ thought they are for everyone; $25 \pm 16\%$ said non-science arts audiences; and one person responded it depended on the aims of the project (see Appendix A for details of statistical techniques used throughout and Appendix B for all the responses). Respondents thought art-science is important because it provides different ways of communicating science, can reach new audiences, can help embed science as part of culture, and that both disciplines can learn from one another through their respective creativity. Therefore Type VI of Malina (2010) was the most highly cited typology of art-science collaboration, though Types IV and V were also mentioned.

There have been numerous published examples of science-inspired artworks (Type V), where science acts as a resource for creative art (Kim, 2011). Voss-Andreae (2011) presents sculptures inspired by quantum physics that he argues can indicate aspects of reality that science cannot. The Tumamoc Hill Arts Initiative was a collection of site-based art and writing inspired by the Sonoran Desert and the underlying science of the region (Mirocha et al., 2015). Similarly, Orfescu (2012) describes artistic interpretations of scientific images, in this instance nanostructures, where artists convert them into pieces of art. Hoare (2013) posits that even classic works of literature, such as 'Moby Dick', have strong scientific influences since art and science were not strictly demarcated at the time. It is therefore clear, even from these few examples, that activities attempting to integrate science into culture are incredibly varied and have been undertaken for a long time.

Engaging with new audiences seems to be a prominent motivation for scientists in undertaking art-science collaborations and many evaluations of art-science events have tried to assess whether they have indeed attracted non-science audiences. Science et Cité, a festival across 20 cities in Switzerland, while striving to be "a festival of the sciences and arts" attracted significantly more people who were interested by science ($40 \pm 1\%$) than art ($24 \pm 1\%$) (respondents could select from any number of 14 options), with the festival's more art-themed events typically only attracting $1.4 \pm 0.3$ times more people with arts interests than science ones (von Roten and Moeschler, 2007). Another example — Covariance, a month-long art-science exhibit in London — found $95 \pm 1\%$ of their audience were frequent or occasional art goers and $83 \pm 3\%$ attended science events, hence there was substantial overlap in these two areas ($\geq 78 \pm 3\%$) (Lynch, 2013). Finally, the Art and Space exhibition in Dunedin, New Zealand attracted audiences $57 \pm 10\%$ of which had a professional background in the arts compared to $26 \pm 9\%$ in science, who primarily attended due to a general interest in art ($71 \pm 9\%$) rather than science ($38 \pm 9\%$) though $50 \pm 10\%$ were attracted by how science and art can combine (Brook, 2017). These case studies therefore highlight that art-science events vary significantly in their audiences' interests and do not necessarily always attract new audiences as desired.

Given the multitude of different formats that constitute the art world, there are many ways of combining it with science. The twentieth century saw film emerge as one of the main art forms readily appreciated by the public (Nowell-Smith, 2017) and in recent years the film festival has burgeoned into an important area of cinema, both culturally and industrially, with an incredibly diverse range of festivals running internationally (Archibald and Miller, 2011). Research into film festival attendees (Báez

and Devesa, 2014) has revealed three key motivating factors: "discovery", "entertainment", and "cinema"; with specialised film festivals (such as those covering specific genres, topics or issues) also providing a general feeling of belonging to a specific group and/or "cinephile community" (de Valck and Loist, 2009; Film Festival Research Network, 2019). Film festivals surrounding science have been growing in number (e.g. European Academy of Science Film, 2019; Imagine Science Films, 2019) with these typically featuring documentary films presenting scientific findings in an entertaining but still educational way. However, beyond simply improving the ways that science is communicated to the public (Type VI of Malina (2010)), there is the potential to as well have science appreciated more as part of culture via film (Type V). BIO-FICTION invites short films addressing current/future debate topics in areas of biology, with a near even mix of fiction and documentary style submissions (Schmidt et al., 2015). CineGlobe is a film festival at CERN which centres around broad and culturally relevant themes inspired by science and technology (CineGlobe, 2019). CineSpace is a film festival by NASA and Houston Cinema Arts Society which solicits films inspired by and using actual NASA imagery.

This paper concerns a film festival project called SSFX (Space Sound Effects), devised and run by the author, which aimed to integrate space science research into culture. The scientific basis for the project was the ultra-low frequency (ULF) analogues of sound present within near-Earth space (e.g. Keiling et al., 2016, and references therein) which had been converted into audible sound (Archer et al., 2018). The motivations for choosing to use these sounds for the creation of art, and in particular through film, are discussed in section 2. The SSFX project had two phases, both with different target audiences and aims. Phase one targeted filmmakers, aiming to engage the independent filmmaking community with the sounds present in the near-Earth space environment and enable the creation of creative short films inspired by and incorporating these sounds. This was tackled by running an international short film competition (adopting standard film festival practises through partnering with film industry professionals) which challenged filmmakers to use the sounds as key creative elements. Section 3 concerns this phase of the project and the subsequent collaborative relationships that formed between scientists and filmmakers through the project. It was through these relationships that phase two of SSFX was possible, which aimed to exhibit these films to wide and diverse audiences, exposing them to this area of space science research and hence positively impacting upon these non-traditional audiences. This phase therefore had two target groups, film exhibitors/programmers and independent film-goers. Section 4 discusses how film exhibitors and programmers were engaged to integrate the films into their events and venues, whereas section 5 concerns evaluating the backgrounds of the audiences that attended these events and what impacts resulted from them.

## 2 Motivations

Space is far from completely empty, it's pervaded with very tenuous plasmas such as the solar wind that streams off of the Sun. Earth's intrinsic magnetic field acts as an obstacle to this wind and results in a magnetosphere, protecting us from much of this harmful ionising radiation. However, the interaction between the solar wind and the magnetosphere is highly complex and dynamic, resulting in phenomena which affect the space- and ground-based technology we increasingly rely upon in modern life such as electrical grids, GPS systems, and weather forecasts. These effects are known as space weather and have been

recognised as a potential risk to our everyday lives (Cannon et al., 2013), however, a large fraction of the general population

are not aware of this (3KQ and Collingwood Environmental Planning, 2015).

One way in which solar wind energy and momentum are transferred into and around magnetospheres are through plasma waves. The spatial and time scales where the weak plasma can largely be treated as a single conducting fluid dictated by magnetohydrodynamics necessitates plasma waves to fall within the ultra-low frequency (ULF) regime, with frequencies fractions of milliHertz up to 1 Hz. Of course this does not lie within the human auditory range, however, simply by dramatically

speeding up playback of satellite observations it is possible to make our ULF wave measurements audible. Archer et al. (2018) converted perturbations in magnetic field data (which move similarly to the plasma itself due to the frozen-in condition within magnetohydrodynamics) taken at geostationary orbit into an audio dataset which is now publicly available from the National Oceanic and Atmospheric Administration (NOAA, 2018). This has already been used as tool in exploratory citizen science with schools, as detailed by the authors, but could lend itself to artistic uses also.

How this audible version of scientific data, or "sounds of space", could potentially be used in the creation of art was primarily informed by the sounds themselves. They surprisingly did not typically take on the musical quality somewhat expected by the researchers who study discrete frequencies and resonances within Earth's magnetosphere, but instead conveyed a sense of dynamism and variety as well as having a somewhat cacophonous nature. While the audible dataset described in Archer et al. (2018) was comprehensive enough in order to undertake science, for the purpose of creating art there was redundancy.

Therefore to reduce the amount of data, only the time-differenced stereo summary files were used, averaging these over all spacecraft to result in only one audio file per year. Through an article in The Conversation (Archer, 2016), republished by Newsweek, Daily Mail, Space.com amongst others, online comments using SoundCloud on what people thought random periods of the data "sounded like" were solicited. While the accuracy and precision of SoundCloud comments' time-tagging meant it could not be used as a means of event identification for scientific research, it did result in the wide range of 85 unique

responses (from 151 comments) that are shown in Figure 1. These reflections on the sounds planted the idea of their usage as sound effects in films, ultimately resulting in the SSFX (Space Sound Effects) Short Film Festival project. In the following sections we detail the various phases and audiences engaged through the project from filmmakers and exhibitors to film-goers, presenting findings on their motivations for getting involved and what impact the project had on them.

## 3   Inspiring creative art-science films

### 3.1   Establishing a film competition

It was clear that even with the idea of having the sounds used in creative ways within films, much expertise and advice from within the film industry was required. We were interested in engaging with the filmmaker community and seeing what they came up with themselves, rather than commissioning something specific from a given filmmaker, therefore we went down the public call model. We solicited expert external contacts from Queen Mary's film department and as well as existing film

contacts. These individuals consulted us on how film festivals operate and are run as well as pointing us in the direction of several organisations and networks that would be helpful. It was deemed that adopting standard film festival practices

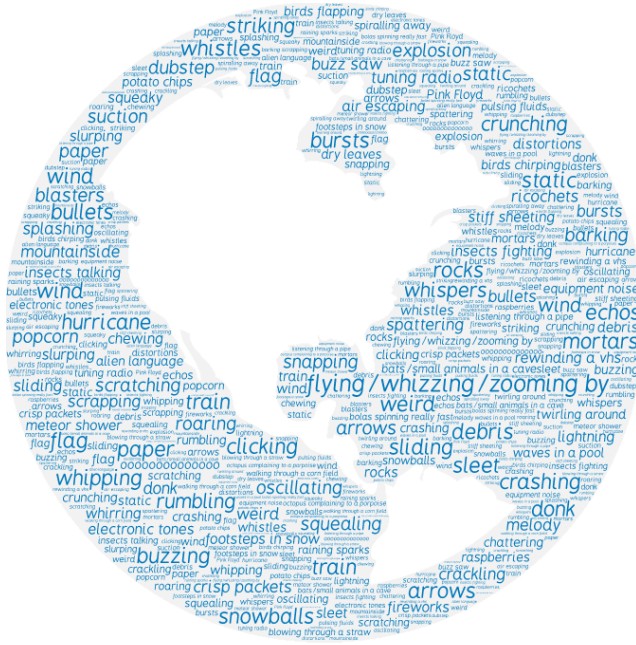

**Figure 1.** Word cloud of comments on what space "sounds like" from SoundCloud playlist of space sounds.

and establishing film industry partnerships were vital in order to make the project as authentic and attractive as possible for independent filmmakers, a concern given this was a new initiative being spearheaded by scientists. However, the film industry professionals we approached to voluntarily sit on the judging panel found the concept exciting:

*"I found the project to be innovative, having never worked with a shorts programme dedicated to engaging film-makers with pre-recorded sounds, space sounds, or an academic programme"* (film industry judge 1)

    *"Many people forget that sound is one of the most important aspects of good filmmaking"* (film industry judge 2)

They were joined on the panel by a couple of scientists with experience in art-science collaborations.

    There were necessarily some differences with the SSFX (Space Sound Effects) Short Film Festival to most film festivals.

Typically these opportunities allow filmmakers to submit existing works with only a few limiting criteria such as genre. However, we were challenging filmmakers to incorporate very specific elements, the provided sounds of space, into their work and in many cases making a film especially for the festival. Given these unusual constraints, it was decided that we would try to make the rest of the competition's criteria as broad and inclusive as possible. Therefore we would not charge a submission fee, there would be no restrictions on genre or topic, we would allow filmmakers to modify the sounds as they saw fit, and permit

films created specifically for the competition or existing films edited to integrate the space sounds. The only other criteria we set were by age and location, with categories initially for both UK and international filmmakers separately in the age ranges: under 18, 18–24, and 25+. The significant work involved on the filmmaker side necessitated there being a large submission

window, which we set as six months long, which would hopefully provide enough time to produce high quality short films. We felt it was important for there to also be monetary prizes associated with the competition to ensure that filmmakers' efforts were valued.

A website was established which hosted the space sounds for download, more information about the competition/festival, and would post YouTube videos throughout the submission window providing more background on the science (SSFX, 2019). However, it was deemed that using an existing online film festival submission platform would be better than coming up with our own method. Desk research highlighted two portals — Withoutabox and Film Freeway (2019). We opted for the former, given it was the first online film festival submission service and was owned by IMDB. In hindsight, however, we realised that Film Freeway would have been more flexible. Withoutabox subsequently closed down in late 2019. While we had set our final submission deadline, staff at Withoutabox recommended within their system that we have various different stages of deadlines ("early bird", "standard" etc.) since this would flag the opportunity to filmmakers looking at Withoutabox's upcoming deadlines calendar. Finally, in order to reduce ineligible entries we asked that filmmakers provide some information on how they used the space sounds. At first this was simply written in the terms and conditions to be included in their cover letter. However, it soon became clear that many filmmakers were not reading the terms and simply submitting their ineligible films anyway. We were able to get Withoutabox to add a custom required question which explicity asked the filmmakers to provide this information, which dramatically cut down (but did not entirely eliminate) spam entries. At this stage the competition was open and we simply needed filmmakers to engage with the opportunity.

## 3.2  Engaging with filmmakers

To share the opportunity widely within the independent film community, existing networks were utilised: a protracted marketing campaign throughout the submission window to Shooting People's over 45,000 member base was run through newsletters, an editorial feature, and social media (Shooting People, 2019); flyers about the competition were mailed to every film school in the country; we attended London-based filmmaker Meetup groups discussing the opportunity with around 70 filmmakers (Meetup, 2019); and we contacted key people recommended by film industry judges for more grassroots marketing. As part of formative evaluation to ensure these were being effective at reaching our target audience, we monitored the number of people who registered interest in the project on our website (essentially subscribing to a mailing list) recording also their age, location, and what their motivations for signing up were. In total 102 people signed up, after having discarded spam entries (see later). The majority of people were 25 or over at $62 \pm 6\%$, with few under 18s at only $10 \pm 3\%$ (in hindsight perhaps to be expected), which informed our merging of the two younger age categories in the competition early in the submission window. In terms of location there was an almost even split in absolute terms between those from London, elsewhere in the UK, and internationally, which is clearly unrepresentative of the global population and likely down to the main networks used to promote the opportunity.

To assess people's motivations we asked them to select from as many of the following options as applied: an existing interest in science generally ($S$), an existing interest in filmmaking ($F$), interest specifically in the project ($P$), or some other reason which they could then specify. We assume that all entries which did not select any of these options (including other) were

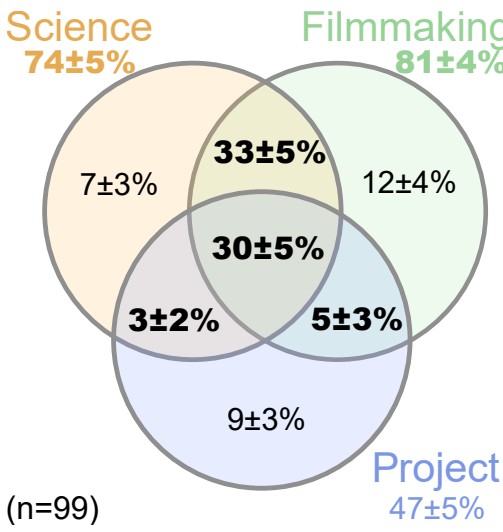

**Figure 2.** Venn diagram of people's reasons for registering interest with SSFX. Bold values denote statistically significant differences from pure randomness (expected 57% in each overall set, 14% in each region) taking into account multiple comparisons ($\alpha_{Bonf} = 0.017$ for sets and $\alpha_{Bonf} = 0.0071$ for regions). Created in part using InteractiVenn (Heberle et al., 2015).

spam. Sixteen people reported other reasons for registering interest (with three not selecting any of the main options): nine had a background in either sound design or musical composition; three were considering visualising the sounds; with others mentioning the creative challenge, interests in space or art-science, and the possibility of integrating the resulting films into existing science or art-science events. Figure 2 shows the breakdown over the three main options. The proportions in each set (i.e. $S$, $F$, and $P$) and region (of the Venn diagram) have been compared to those expected purely at random, with $S$, $F$, and $S \cap F$ being significantly greater than expected. We also compared the sizes of all sets and unions of sets with one another finding that most of these differences are statistically significant — of the 6 possible comparisons only $S$ vs. $F$ and $F$ vs. $S \cup P$ were not. From this we deduce that people who registered interest typically had existing interests in both science and filmmaking. Given only small (typically only a few percent) fractions of the public work or have qualifications in film (e.g. BFI, 2019), we conclude that SSFX successfully engaged the filmmaking community. Given the considerable effort involved in creating a film, an interest in science also is thus understandable, though anecdotally from conversations with filmmakers it was found that their primary scientific interests (if any) were typically not in physics or space science though.

Leading up to the competition's deadline very few films had been submitted, therefore to assess this we sent out a survey to the mailing list six weeks before the deadline. While only seven responses were received, six indicated they intended to submit films though only two of these were confident that their film would be ready by the deadline. Based on these results we decided to extend the deadline by an additional two weeks to allow more time for filmmakers, while still having sufficient time for judging and event organisation. Even closer to the submission, a few filmmakers reached out stating that their films

would not be complete so we took the decision to allow work in progress submissions, so long as the filmmakers indicated what additional work needed doing and that it could be achieved in time for the event.

## 3.3 Evaluating film submissions

By the deadline 22 eligible films had been submitted (180 ineligible films not featuring any space sounds were also submitted, most of which came before the bespoke question was implemented), which according to their credits involved a total of 90 people. These films themselves demonstrate an impact on filmmakers, given that they have engaged with an opportunity to co-create an art work based around scientific data — a substantial undertaking. Most entries (nine) were in the 25+ UK category with 4–5 entries in the other three categories. This also meant that most entries came from the UK ($59 \pm 12\%$) though we note international filmmakers from Brazil, Canada, Italy, Portugal, and USA also submitted films. None of the differences in submission numbers were statistically significant by category, age, or internationality.

Each film was scored by the judges on both their usage of the space sounds (e.g. a few submissions had just a token usage of the sounds within their films) and overall impression of the film with equal weighting within Withoutabox's online system. Judges could also leave any written feedback on both judging criteria to help final decisions. A subset of all the submissions based on total runtime were assigned to each judge, though 11 films were seen by all judges (the shortest ones) and at least 3 judges saw each submission. One of the film industry judges noted:

> *"The process of running the competition was extremely professional and I would recommend the model to others in the future, with a secure screening system, showing full creative credits for each film that allowed feedback to be added and votes cast within one dedicated site. Thanks for such great organisation and clear steer on how you wanted the judging to go."* (film industry judge 1)

In the end there was a fair amount of disagreement between the judges — the alpha coefficient of Krippendorff (1970, 2018) for these ordinal measurements was only $0.43$ (where a value of 1 would indicate perfect agreement and 0 would result from randomly drawn scores, see Appendix A). Each judge's scores were therefore standardised, using means and standard deviations across only those films which were seen by all, to ensure no one judge had more sway in the outcome. Given the overall time within the venue we were able to select the top eight films (based on the average standardised scores) for exhibition, however, one of these was unfinished upon submission and could not ultimately be completed in time. This film dropping out necessitated merging the two international categories.

One of the film industry judges noted about the submissions

> *"I was really impressed by the quality and diversity of films submitted through the competition as well as the international uptake. The range of film making styles was really interesting, there were dramas, comedies, animation, science fiction and avant-garde productions with some films exploring the scientific concepts directly and others using them in more abstract ways... I really loved the fact that the project was open in how filmmakers could interpret the sounds in their productions and I think this was key to gaining the variety that appeared across the submissions."* (film industry judge 2)

We note that even while some of the films incorporated elements of the underlying scientific concepts, they were not the primary focus of the films and were done in creative fictional ways and not in a documentary style. Thus all the films fell under the "science as culture" type of art-science, i.e. Type V of Malina (2010), rather than a form of direct science communication (Type VI). Even one of the judges noted "*it has genuinely got me thinking about how I could explore some of the research in my future creative outputs*" (film industry judge 2). The conclusion from these is that by giving a huge amount of creative freedom over to the filmmakers in allowing them to interpret and include this scientific data as they saw fit, it enabled the creativity and variety of films submitted, thus highly aligning with the aim of integrating science into culture. The following sections summarise the selected works, including the filmmakers' reflections captured during filmed panel discussions at events.

### 3.3.1 Astroturf

Synopsis: A meticulous young man tends to his fake garden to the sounds of deep space
Genre: Science Fiction
Duration: 1 min

The film depicts a man performing gardening tasks, though this garden features no real plants instead being filled with the titular astroturf along with plastic flowers, trees, butterflies etc. It is revealed at the end that this garden in on the moon and that the Earth is on fire. The director noted

> "*We wanted to make a film that used the space sound effects in an interesting way, while telling a compelling short science-fiction story. The rustling, swirling space sounds reminded us of the noises that people make all the time when performing simple tasks - sounds that in film are often replaced or reproduced as foley. So we decided to build the entire soundtrack from the space sound effects, and created a simple narrative that involved a combination of actions that we felt would be convincing when dubbed. We came up with the idea of putting it in space because of the space sounds... Because [in the film] we've screwed up the Earth, for him this tiny patch of land is extraordinarily precious and so that was where his character emerged from.*" (director of 'Astroturf')

The producer added

> "*He's trying to recreate what he's known in the past, what a real garden is, in this fake world that he's living it. Because we're both 'greenies' and we see that as a potential future for us, that was what really inspired that incongruess nature of nature versus fakeness.*" (producer of 'Astroturf')

Further comments can be found at https://www.youtube.com/watch?v=RpC-sFzUnEE.

### 3.3.2 Dark Matter(s)

Synopsis: An experimental and meditative imagining capturing the activities of a fish tank in a way that takes the inhabitants out of their enclosed world, to a place unknown, to feel both their death and their life.
Genre: Video Art
Duration: 5 min

The director describes the film as

> *"about a couple fish in a fish tank, but we tried to film it in a way that it doesn't look like they're in a fish tank...*
> *that got rid of the boxed in enclosure. When I realised what's bigger than being a locked in a box was everything,*
> *it made such sense to me to look into sounds from space. I think the sound from space gives it that extra push for*
> *[the fish] to like break out of this cage or for that dichotomy of inside versus outside to be transcendent. As soon*
> *as I figured out that I was going to use sound effects from space I think the project came full circle."* (director of
> 'Dark Matter(s)')

The sound designer commented on how he modified the sounds for the film, which were matched with classical music throughout:

> *"I used a lot of reverbs to soften them [the sounds], I did a little bit of slow panning, I shifted the pitch on the*
> *sound effects a couple times to separate them, I shifted them down to make them a little deeper. There's this one*
> *part where there's a bunch of bubbles and so I changed them so they could sound like bubbles. Basically the whole*
> *process was making them soft enough to fit in the context of the film."* (sound designer of 'Dark Matter(s)')

Further comments can be found at https://www.youtube.com/watch?v=quLaFmS9kDE.

### 3.3.3 Murmurs of a Macrocosm

Synopsis: A journey through a microscopic world. We are led via the descriptive recordings of those who travelled it.

Genre: Science Fiction

Duration: 5 min

This film shows recoloured drone footage from Snowdonia paired with NASA Apollo recordings and the space sounds to depict exploration in a microscopic realm, which is revealed at the end to be inside the grooves of a vinyl record of the moon landings. The filmmaker stated

> *"It was a visual that I always loved to, those SEM microscopic images that are colourised black and white images.*
> *But I always wanted a little bit more, I wanted to move around them. I think when hearing those sounds it kind*
> *of reminded me a bit of a record player. It also reminded me a lot of the sound from space in 'Contact', the Carl*
> *Sagan film / book, that's how it came together from those little things."* (director of 'Murmurs of a Macrocosm')

When discussing the use of NASA recordings he noted

> *"finding this sort of innate humour and human conversation that they would often have to each other and they just*
> *felt so in awe the whole time... they're constantly excited which I really I love that aspect of exploration."* (director
> of 'Murmurs of a Macrocosm')

Further comments can be found at https://www.youtube.com/watch?v=e3kUGlvI_Hk.

### 3.3.4 Names and Numbers

Synopsis: A sound and voice collage shaped by the sounds of space and Morse code, addressing the external, physical and material experiences of sound and movement contrasted with interior reflections, explored through language, inner voices and symbols.

Genre: Experimental

Duration: 14 min

The filmmaker explained how this experimental piece came together:

> *"I tried to enact that experiment of writing down your thoughts to the sound of a buzzer which samples your mind at any particular time... It was basically an accumulation of ideas and just sitting down and following the logic of each individual material thing: a soundtrack, recording a piece of text, a collection of different images. There was no simple way of putting them all together and I guess the stream of consciousness of that writing process was one of the guiding principles."* (director of 'Names and Numbers')

Further comments can be found at https://www.youtube.com/watch?v=Uuvcm1YfdZ4.

### 3.3.5 Noise

Synopsis: A secretive woman opens herself up to her unruly housemate, after they are stuck together in her room.

Genre: Drama

Duration: 13 min

This drama film is about a woman who often isolates herself by listening to the sounds of space, and who doesn't get along with her very different housemate. They eventually are able to connect over these sounds. The director noted on the title

> *"noise is a specific scientific term for something which has no informational value... and so when the characters are talking to each other they're trying to work out what's noise and whether they can actually understand anything from what they're saying to each other... Once you heard the sounds they kind of wrote the story, they had to carry the narrative, creating a character in and of themselves."* (director of 'Noise')

Further comments can be found at https://www.youtube.com/watch?v=Fgvo_lP7ZmA.

### 3.3.6 Saturation

Synopsis: There's no answer when time is the question. Featuring 35mm slides found in a medical archive, this sci-fi story concerns unknown phenomena that made all organic processes so fast as to make life impossible.

Genre: Science Fiction

Duration: 7 min

This film couples still images of medical imagery with subtitled text and a soundtrack composed of modified space sounds. The filmmaker explained the creative process

*"When I first started to edit the film a couple of years ago, before I even knew what it was supposed to be, I thought [the mysterious phenomenon in the film's narrative] was something related to space... When I saw your call [for films using the sounds of space], I realised that's what I needed - something really from space that I can use on my film. To make the sounds more tense I saturated them, making them more drone-like... I was also interested by the process itself of making the sounds hearable by stretching and compressing the time and it is very related to*

*the narrative that I was thinking.*" (director of 'Saturation')

Further comments can be found at https://www.youtube.com/watch?v=rYxFHExQ4aQ.

### 3.3.7 The Rebound Effect

Synopsis: Bringing together contemporary movement and digital media to capture dance in a way which pushes beyond the tangible dimensions of live performance.

Genre: Dance / Music Video

Duration: 2 min

This film depicts a modern dancer moving to the sounds from space mixed with electronic music. Unfortunately filmmaker comments about this work were not captured.

## 3.4    Running a film festival

A boutique arts and cinema venue was hired for the SSFX Short Film Festival (Rich Mix in Shoreditch, London). To capitalise on their regular members, we opted to have them host ticket sales and primarily undertake marketing for the event. While the event was not being run to make a profit, we decided to charge a small ticket price to reduce cancellations and convey a sense of perceived value for the event. In reality all ticket proceeds actually went into the cost of a free post-event reception. The cinema required the films in a digital cinema package (DCP) format. First we received the high-quality video files from the

selected filmmakers and then converted these using the free open-source DCP-o-matic (2019) software. For exhibition to the public the films required British Board of Film Classification (BBFC) certification also, which were submitted as DCPs online. The majority of classifications were Universal rating bar two — one was deemed Parental Guidance for "mild surgical detail" and another gained at 15 rating due to "strong language and drug misuse".

The event was started with an unadvertised short presentation on the underlying science to the audience. These were then

followed by groups of film screenings, awards presentations, and panel discussions between scientists and filmmakers (international filmmakers joined via video conferencing) about their work and approach to the project. Photos of all of these can be seen in Figure 3. The post-event reception then enabled further discussion between scientists, filmmakers, and film-goers. Evaluation of this event (and subsequent ones) can be found in sections 4 and 5.

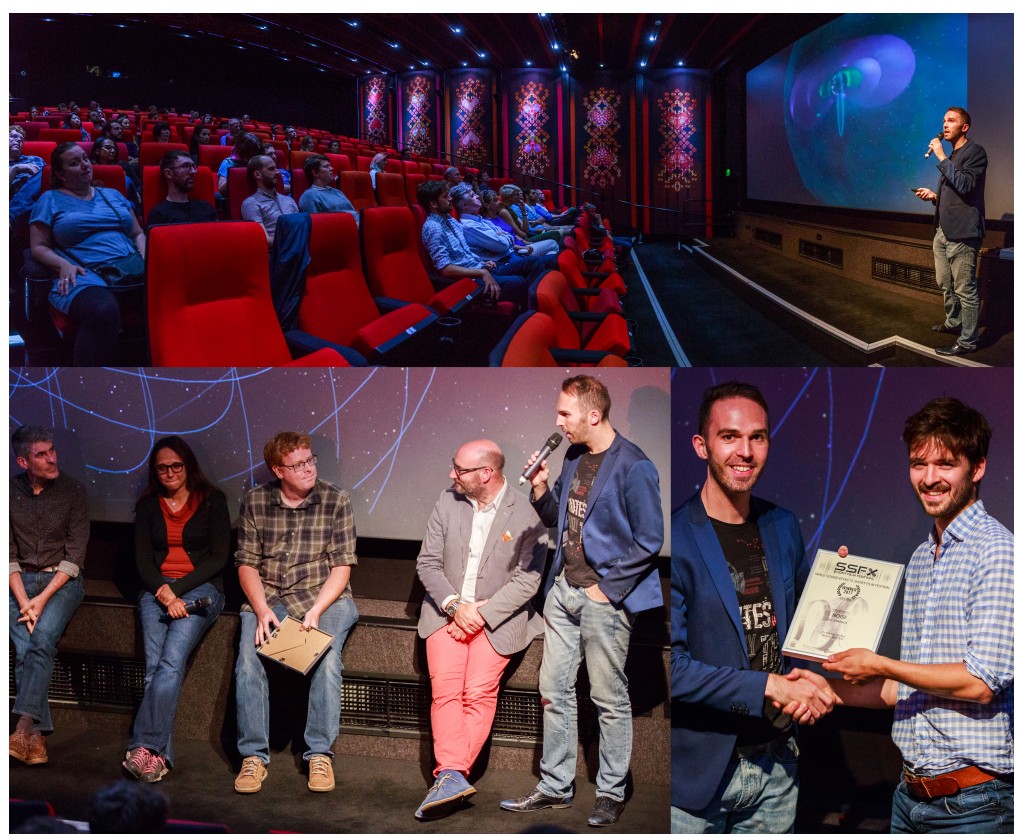

**Figure 3.** Photos from the SSFX Short Film Festival. (Top) Presentation about the underlying science. (Bottom Left) Panel discussions between scientists and filmmakers. (Bottom Right) Awarding of prizes to filmmakers.

## 3.5 Supporting the films and filmmakers

Following the SSFX Short Film Festival, we wanted to support the filmmakers in sharing their work more widely. In return we asked them to add specific prologue/epilogue text about the underlying science as well as items in the credits pertaining to project staff, data providers, and funders. In hindsight, it may have been easier to ask for this at the selection stage so these would have been incorporated into the high quality versions provided for the festival.

There were a number of different ways in which we supported the filmmakers. Firstly at the level of individual films we financed the submission fees for the top four highest scoring films ('Astroturf', 'Dark Matter(s)', 'Murmurs of a Macrocosm', 'Noise') to existing UK film festivals, as this was flagged by the filmmakers as a limiting factor in their ability to share the work more widely. We left it up to the filmmakers to determine which festivals might be the best fit for them given the budget offered to each. Secondly, we acted as champions representing the entire set of shorts, approached numerous film exhibitors to enquire about some of them being considered for screening within their existing events. Finally, we wanted to offer the entire set of shorts as a ready-made package that could be screened elsewhere. However, it was deemed that simply showing the

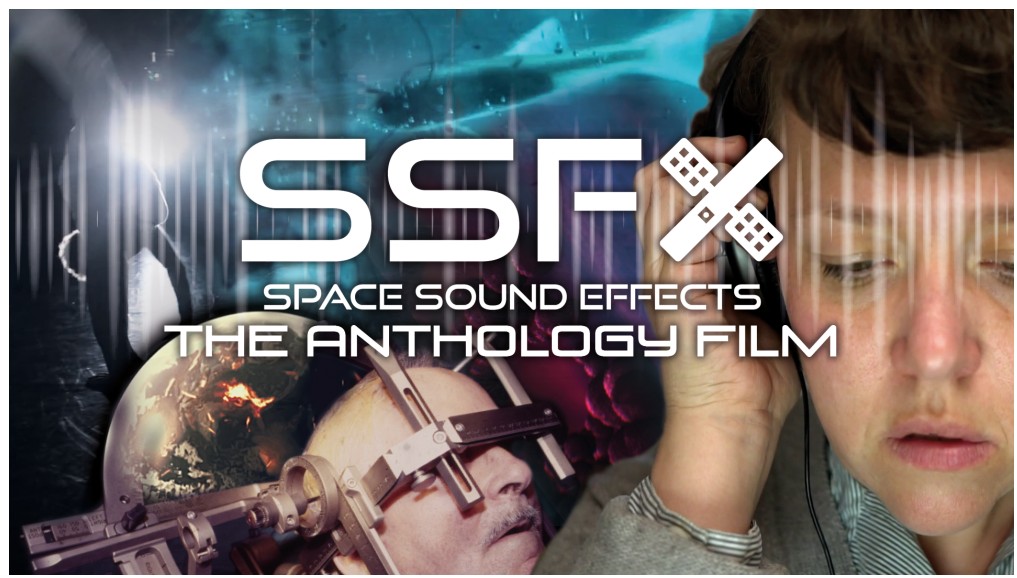

**Figure 4.** Anthology film poster composed of imagery from each of the selected shorts.

shorts back-to-back and not also having some background to their context and production would not be such an informative nor entertaining experience for audiences. Therefore, it was decided to produce a framing film which would incorporate the shorts into an anthology.

A tender was put out to various production houses and through networks soliciting pitches for this framing film. The aim was that this framing film could in a narrative way bridge together the short films while also communicating a few simple messages about the underlying science behind the space sounds and how they affect us. We received three proposals, which were very different in approach. We went ahead with one which envisioned a point-of-view shot film during a space weather event, where technology failing causes the films to appear on various screens around the house. We worked closely with the production company on the development of the script and through the production process. The poster for the anthology film, composed of imagery from the individual short films, is shown in Figure 4. This anthology was released for free on YouTube in October 2018 (https://www.youtube.com/watch?v=P5_OljSnA1k).

Overall, the filmmakers were very grateful for all the support offered. One of the filmmakers noted:

> *"The SSFX project was incredibly rewarding and allowed us as a creative team to learn about an exciting area we had no knowledge of previously. We also met other interesting filmmakers and found the audiences incredibly engaged and interested into the background of the project and where the research is. We have had a lot of positive feedback and have been able to direct our audiences to the SSFX website for more information. Martin Archer has been incredibly supportive and a champion for these films, for which we are incredibly grateful and have found invaluable."* (producer of 'Astroturf')

Another said:

*"Collaborating on the project was a wonderful experience and we were so grateful for all the opportunities offered to our film from taking part, reaching an international audience with our film and getting to enter into dialogues with audiences, scientists and other filmmakers. If it wasn't for SSFX, I'm not sure that our production team would have thought to engage so thoughtfully with sounds from space – we didn't know they existed. Not only did Martin provide exceptionally unique and compelling sounds for any sound designer to work with, but he was so thoughtful in terms of explaining the science and techniques used to capture these sounds. As someone who was a little intimidated by science in school, I really felt an understanding around the basic mechanisms and significance around obtaining these compelling sounds. As a filmmaker, I appreciated this so that I could answer questions at Q and As with confidence and ease. One of the other aspects about this project that I appreciate was Martin's ability to have people from all over the world join in on the conversation. Multiple times throughout this process, I was able to talk to audience members and fellow filmmakers on a different continent while staying in the US. This is not something that I've experienced a lot in the independent film festival world. If you can't physically attend, then you simply can't be a part of the conversation. That was not the case with SSFX and it made the experience all the more educational, inclusive and fun for everyone involved."* (director of 'Dark Matter(s)')

These comments highlight the impact that the open, collaborative and supportive approach that SSFX took had on the filmmakers.

## 4 Infiltrating film events

Several complementary approaches were taken to get the SSFX films more widely seen as part of film festival and events programmes. As noted earlier, we paid the submissions fees for film festivals (limited to the UK only due to funding usage restrictions) identified by the filmmakers. In addition to this, we advertised the free shorts and/or anthology through film exhibitor networks recommended by our judges (including the various BFI Film Audience Network hubs and the Independent Cinema Office) and approached several key film exhibitors, enquiring about either arranging one-off screenings of the anthology or showing the shorts before their features. This generally fed into their aims of developing audiences for and increasing access to a diversity of film content for local independent film-going communities. We also liaised with a few science focused events such as science festivals, either through open calls or those that approached us, about integrating the films into their programme in some way.

Table 1 details all the events which featured SSFX film screenings, where these have been grouped by initiative since in several cases multiple screenings of the same or different films occurred. There was a large overrepresentation of UK-based events ($68 \pm 10\%$) compared to all film festivals globally ($8\%$ , $p = 3.5 \times 10^{-17}$) as listed in Film Freeway (2019). This was in part due to funding usage restrictions limiting which festivals could be applied for, however, we also note that many film festivals aim to highlight the works of filmmakers from their own country.

Events have been classified as either art, art-science, or science, with the distribution of event types shown in Figure 5. Art events denote those with no clear association with science whatsoever, science events indicate those with no explicit link

| | Event | Location | Type | Pre-existed | Facilitator | Shown | # Screenings | Audience |
|---|---|---|---|---|---|---|---|---|
| a | Academia Film Olomouc | Olomouc, Czech Republic | Art-Science | Yes | Filmmaker | Shorts (#3) | 1 | 70 |
| b | Aesthetica Short Film Festival | York, UK | Art | Yes | Filmmaker | Shorts (#3) | 2 | 71 |
| c | AM Egypt Film Festival | Cairo, Egypt | Art | Yes | Filmmaker | Shorts (#2) | 1 | 100* |
| d | British Science Festival | Hull, UK | Science | Yes | Scientist | Anthology | 1 | 25 |
| e | Cambridge Film Festival | Cambridge, UK | Art | Yes | Filmmaker | Shorts (#1) | 1 | 100* |
| f | Central Film School | London, UK | Art-Science | No | Scientist | Anthology | 1 | 10 |
| g | Escape Velocity | Maryland, USA | Art | Yes | Filmmaker | Shorts (#3, 5) | 2 | 100* |
| h | Genesis Cinema | London, UK | Art-Science | No | Scientist | Anthology | 1 | 25 |
| i | Grand Concourse Film Screening Series | New York, USA | Art | Yes | Filmmaker | Shorts (#2) | 1 | 100* |
| j | Imagine Science Film Festival | New York, USA | Art-Science | Yes | Filmmaker | Shorts (#1) | 1 | 51 |
| k | INDUSTIREAL Video Art | Oradea, Romania | Art | Yes | Filmmaker | Shorts (#2) | 1 | 100* |
| l | Les Films de la Toile | Paris, France | Art | Yes | Filmmaker | Shorts (#2) | 1 | 100 |
| m | Liverpool Film Festival | Liverpool, UK | Art | Yes | Filmmaker | Shorts (#3) | 1 | 100* |
| n | London Short Film Festival | London, UK | Art | Yes | Filmmaker | Shorts (#1) | 1 | 75 |
| o | Nightstar Cinema | London & Dorset, UK | Art | Yes | Scientist | Shorts (#1, 2, 3, 7) | 4 | 333 |
| p | Nozstock: The Hidden Valley Festival | Hereford, UK | Art | Yes | Scientist | Anthology | 1 | 25 |
| q | On The Other Side From You, East End Film Festival | London, UK | Art | Yes | Filmmaker | Shorts (#3) | 1 | 102 |
| r | Rooftop Film Club | London, UK | Art | Yes | Scientist | Shorts (#3) | 1 | 155 |
| s | Royal Society Summer Science Exhibition | London, UK | Science | Yes | Scientist | Shorts (#1, 2, 3, 5, 7) | 5 | 70 |
| t | SCInema International Science Film Festival | Various, Australia | Art-Science | Yes | Filmmaker | Shorts (#1) | 6 | 1,800 |
| u | SCInema Community Screenings | Various, Australia | Art-Science | Yes | Filmmaker | Shorts (#1) | 818 | 89,000 |
| v | Shorts on Tap | London, UK | Art | Yes | Filmmaker | Shorts (#1) | 1 | 120 |
| w | SMASHfest | London, UK | Science | Yes | Scientist | Shorts (#1, 2, 3, 4, 7) | Exhibition | 2,076 |
| x | South London Shorts | London, UK | Art | Yes | Scientist | Shorts (#3, 7) | 2 | 80* |
| y | Southampton International Film Festival | Southampton, UK | Art | Yes | Filmmaker | Shorts (#1) | 1 | 15 |
| z | Space - Music & Film inspired by the cosmos | London, UK | Art-Science | Yes | Scientist | Shorts (#1) | 1 | 40 |
| aa | Space Lates | Leicester, UK | Science | Yes | Scientist | Shorts (#1, 2, 3, 7) | 4 | 200 |
| ab | SSFX Short Film Festival | London, UK | Art-Science | No | Scientist | Shorts (#1, 2, 3, 4, 5, 6, 7) | 7 | 85 |
| ac | SSFX Anthology Premiere | London, UK | Art-Science | No | Scientist | Anthology | 1 | 35 |
| ad | Storyhouse | Chester, UK | Art-Science | No | Scientist | Anthology | 1 | 10 |
| ae | Viten Film Festival | Bergen, Norway | Art-Science | Yes | Filmmaker | Shorts (#6) | 1 | 40 |

**Table 1.** List of SSFX events. Individual shorts are numbered as per section 3.3. Asterisks (*) denote estimated audience figures due to lack of information from event organisers.

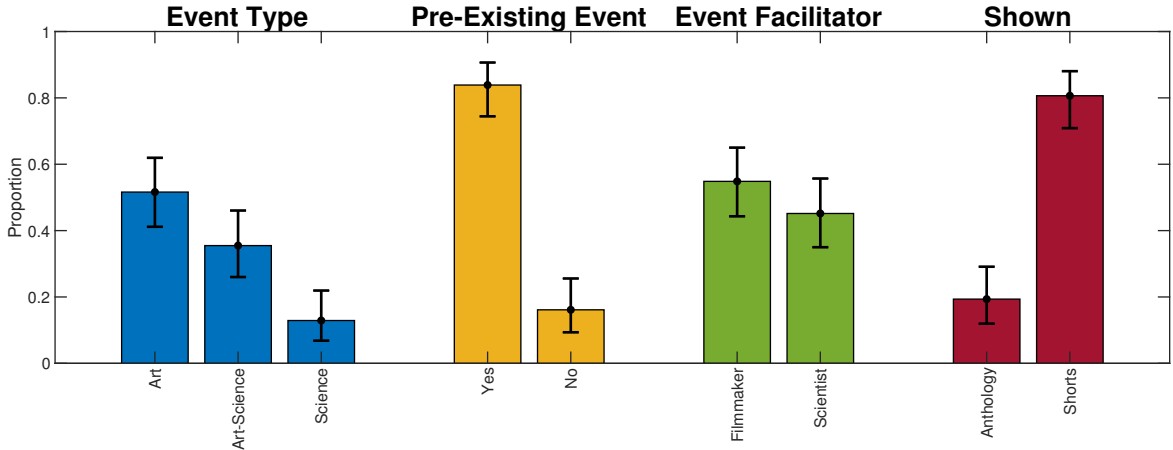

**Figure 5.** Summary statistics of SSFX events.

to art, and art-science is used to describe events with a stated connection between the two subjects. There were substantially more art events than science ones ($p = 0.012$, $\alpha_{Bonf} = 0.017$), which constituted the only significant difference between SSFX event types. Excluding science events, art-science ($41 \pm 11\%$) was overrepresented compared to all film festivals globally that contain some mention of space or science ($11\%$, $p = 6.1 \times 10^{-5}$). Both of these results reflect some of the struggles faced in the second phase of the SSFX project — science event programmers were largely uninterested in art-science since their audiences are already highly engaged with science and not necessarily with art, whereas many film event programmers we approached struggled to understand the concept of the project thinking the films were aimed at science audiences rather being open to judge them as films in their own right that happen to contain a scientific connection (i.e. their preconception was Type VI of Malina (2010) rather than Type V).

Figure 5 demonstrates that screenings predominantly occurred as part of pre-existing events rather than at bespoke ones ($p = 1.9 \times 10^{-4}$), indicating SSFX was largely successful at infiltrating science into the film world, and there was a fairly even split in events arranged by filmmakers or scientists. We note that filmmakers were more successful at infiltrating art events ($71 \pm 14\%$ of all their events) than the scientist ($29 \pm 15\%$), though this difference was not strictly statistically significant when accounting for multiple comparisons ($p = 0.021$, $\alpha_{Bonf} = 0.017$). Primarily it was an individual short film or subset of the collection of shorts which was exhibited at events rather than the full anthology film ($p = 8.8 \times 10^{-4}$), which we struggled to convince film programmers to incorporate into events despite advice from film industry collaborators that this might be an attractive proposition. Of the individual shorts 'Astroturf' was the most successful, though the only statistically significant differences ($\alpha_{Bonf} = 0.0024$) in the number of distinct events/initiatives by film were between 'Astroturf' and both 'Names and Numbers' ($p = 5.0 \times 10^{-4}$) and 'Saturation' ($p = 1.9 \times 10^{-4}$). We note that neither of these latter two films' festival submission fees were funded by the project and in the case of 'Saturation' a number of exhibitors expressed that they could not screen it at their family-friendly events due to the potentially upsetting medical imagery (edited clips from 'Noise' removing the strong language and drug usage were however able to be used). In terms of total number of screenings, 'Astroturf' had

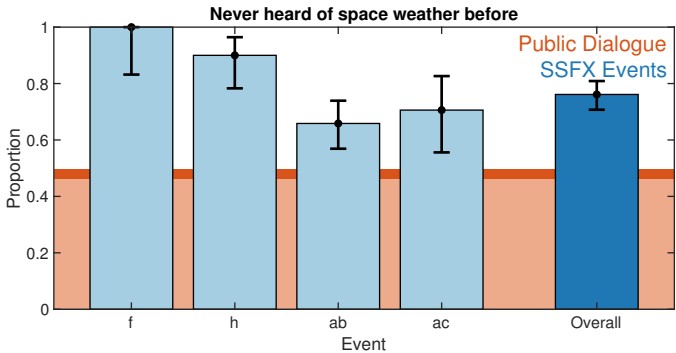

**Figure 6.** Prior knowlege of audience at SSFX art-science events in blue compared to a recent public dialogue (3KQ and Collingwood Environmental Planning, 2015) in orange.

significantly more than all the other shorts though this was purely due to being selected for the touring SCInema International Science Film Festival (and associated community screenings) across Australia.

Film festival acceptance rates are typically $\sim 5\%$ (Stephen Follows, 2013b) with the largest festivals being $\lesssim 1\%$ (Sponring and Puskás, 2018). While we do not have concrete numbers on exactly how many festivals the filmmakers submitted to, given the budget and average submission cost for short films (Stephen Follows, 2013a) we estimate around 30 total submissions. This means that the 17 festival successes constitutes an impressive acceptance rate across the shorts of $57 \pm 11\%$, significantly higher than expected. This perhaps reflects the quality of the art-science films that resulted from the project. We also note that given the filmmakers were submitting their shorts to festivals independently and all found success, this lends confidence beyond just an individual case study that this model of infiltrating science into cultural events can indeed work.

## 5 Engaging audiences through film

We generally relied on the event organisers to attract audiences, since they have built-in audience bases from their previous activities. Given we were largely infiltrating existing events, this limited the evaluations that could be implemented especially as at many events (especially the international ones) no filmmakers or scientists from the project were physically present. Therefore, evaluation data was typically collected only at bespoke SSFX events and several methods were employed: a ball and bin question upon arrival assessing prior knowledge, graffiti walls at post-film receptions assessing their motivations and takeaways, and an online survey three weeks later for those who left contact details. Filming by a third party at the SSFX Short Film Festival (event ab) captured additional qualitative data. Given that these events where evaluation was possible tended to show all the shorts (either individually or via the anthology) we are unable to comment on whether certain SSFX films were more impactful upon attendees than others.

As part of a recent public dialogue, 3KQ and Collingwood Environmental Planning (2015) found in a survey of 1,010 people representative of the UK adult population (by gender, age, social grade, education, dependants, geographic region, and human

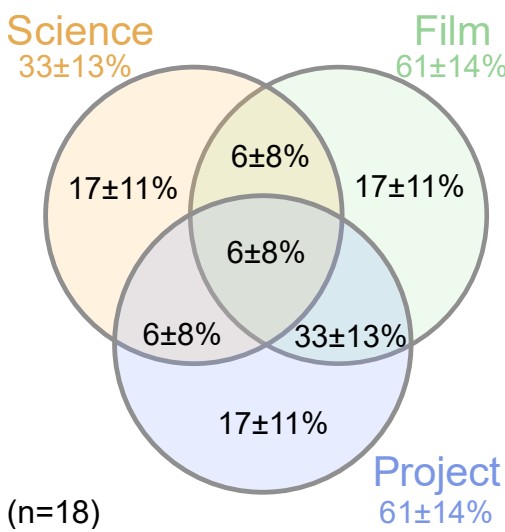

**Figure 7.** Venn diagram of people's motivations for attending SSFX art-science events in the same format as Figure 2.

settlement type) that $48 \pm 2\%$ have never heard of space weather before. We therefore asked audiences upon arrival at events whether they had heard of space weather before, via a ball and bin method where attendees were instructed to put a ball in either the 'yes' or 'no' bin. The results from the individual events where we asked this question are displayed as the light blue bars in Figure 6 indicating levels greater than in the public dialogue (orange). Combining the data from all these SSFX events gives an overall result (dark blue) of $76 \pm 5\%$, which constitutes $2.95 \pm 0.20$ times more likely (the odds ratio) to have never heard of space weather than the general population ($p = 9.2 \times 10^{-8}$). Therefore an atypical audience was attracted to these events in terms of prior knowledge. Note that these results came exclusively from art-science events and arguably one might expect an even greater overall proportion of people to be unaware of the field at the art events that SSFX infiltrated.

Another way we assessed whether the project attracted new audiences was by asking what motivated them to attend. At two art-science events (ab and ac) this was collected via open-ended graffiti walls, where 9 responses were recorded which can be found in Appendix B. Through thematic analysis (Braun and Clarke, 2006) it was possible to group all of these as being due to an interest in science (e.g. *"love science"*), film (e.g. *"I like weird films"*), or specifically the project (e.g. *"interesting concept"*), where the quotes displayed serve as representative illustrative examples from different respondents. Follow-up online surveys after several events (f, h, aa, ab, and ac) specifically asked in a closed question whether attendees had been attracted due to regular attendance at science events, film events, or if it was specifically this event that had interested them. They could select as many options as were applicable. Given this yielded only 12 responses we opt to combine the data from both methods, omitting event aa since out of those events surveyed it was the only science event as well as the only pre-existing one. The overall results are shown in Figure 7. Repeating the same analysis as with filmmakers' motivations revealed that significantly more people attended due to being film-goers or specifically being interested in the project ($F \cup P$) at $78 \pm 12\%$ compared to attending science events often ($S$) at $33 \pm 13\%$ ($p = 0.0023$, $\alpha_{Bonf} = 0.0083$). This therefore provides

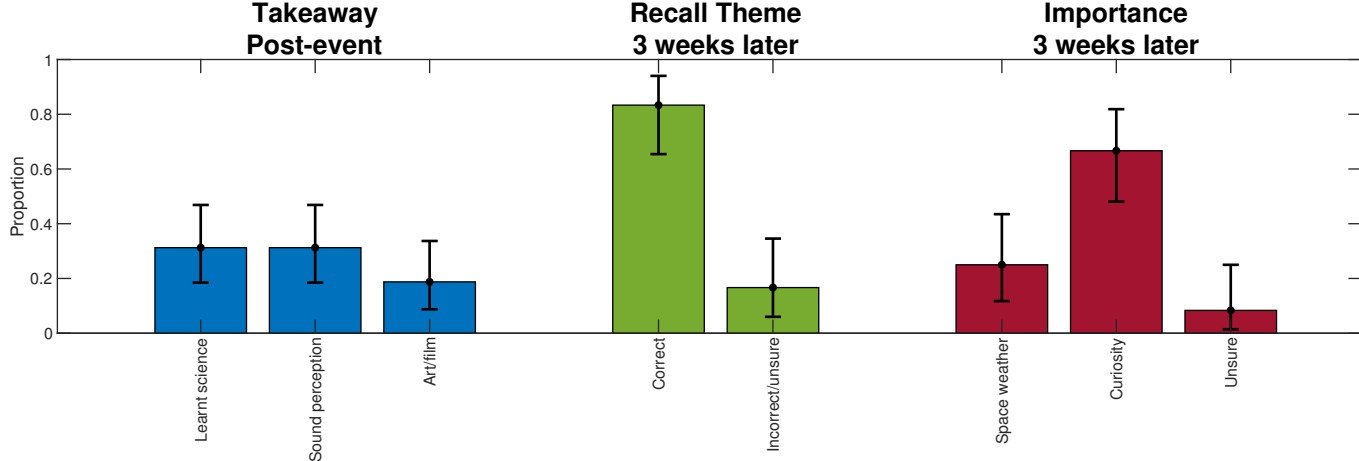

**Figure 8.** Summary statistics of evaluation data following SSFX events.

further evidence that SSFX was able to attract substantial non-science audiences, placing it as comparable to some of the most successful art-science events across different art forms at reaching new audiences (cf. Brook, 2017). Again we note that since this analysis pertained only to art-science events, it is highly likely at the art events SSFX infiltrated that even fewer people would have exhibited science interests given the complete lack of a science-connection at these events and the existing research into the motivations behind film festival attendance (Báez and Devesa, 2014).

We assessed the learning outcomes of attendees through the follow-up surveys, asking in open questions if they recalled the event's theme and why this topic is studied/important. The full list of responses is given in Appendix B. As shown in Figure 8, the majority ($83 \pm 14\%$, $p = 0.039$) correctly recalled that the event concerned the sounds of space or provided more specific answers. Interestingly most ($67 \pm 17\%$) thought this area was important due to the inherent value of science / intellectual curiosity (e.g. *"It helps us to understand the universe, physics, and gives us a clearer idea of the world around us"*) rather than citing space weather ($25 \pm 16\%$), though this majority was not statistically significant and neither were the differences between responses. As far as we are aware there is little published research into the recollection of public engagement events' themes and key messages by attendees in follow-up surveys. However, comparing with studies into the recollection of television campaigns (e.g. Berry et al., 2009; Potter et al., 2019) and the so-called Memory Chain Model (Murre and Dros, 2015) suggests that the fraction quoting space weather would be deemed successful, while the recollection of the event's overall theme would be considered extremely high. Also, given the atypical non-science audience, the fact that many attendees took away from the event the value of fundamental scientific research was an unanticipated but very welcome impact.

In terms of impact on attendees, at two art-science events (ab and ac) we asked via graffiti wall what (if anything) they had gained or taken away from the event. Most of the 16 responses, displayed in full in Appendix B with representative examples from different respondents shown here, could be broadly categorised using grounded theory (Silverman, 2010; Robson, 2011) as concerning the science (e.g. *"amazing space sounds I want to learn more about"*), perception of sound (e.g. *"people hear differently"*), or art/film (e.g. *"grown an interest in film-making"*), with the proportions of each shown in Figure 8

demonstrating a near even split between the three. Other miscellaneous takeaways included aspects of science communication, humanity, and specific (non-science) themes raised in the films. Furthermore, at the SSFX Short Film Festival (event ab) a selection of people were interviewed during the reception by a third party, with some of the responses available at https://www.youtube.com/watch?v=sPa7avaksFI. The most common point that emerged (again analysed using grounded theory) was that attendees really enjoyed the broad range of interpretations of the same space sounds which were expressed in the different films. Others commented on how the concept of the festival was an interesting approach of bringing scientific ideas to a wider audience, that they had learned about and gained an interest in the science behind the sounds, and that it attracted a diverse group of people with a lot of interaction particularly in the reception. On this latter point, it was anecdotally noted at most of the events that the diversity of audiences by gender and ethnicity appeared much greater than compared to typical physics engagement events, though this was not captured quantitatively. Respondents in the follow-up survey also noted other takeaways from attending which we again have coded using grounded theory and provide illustrative example responses from different participants here: enjoying or being inspired by the event (e.g. *"Really enjoyed the enthusiasm of the speaker and the topic of the films mixed with science"*), the creativity/diversity of films (e.g. *"how each filmmaker found the humanity in sounds from space"*), meeting and hearing from both scientists and filmmakers (e.g. *"It was interesting to meet some of the people involved in both the science and filmmaking"*), and the importance/relevance of the scientific research (e.g. *"Genuine and relevant science research and knowledge is vital and underused in the film industry"*). Of the responses which did not fit into any of these themes, one person said that they had developed an interest in arts events by attending (*"I will definitely look at the* [arts venue] *Rich Mix website more for future events"*) while another found the anthology film to be *"very strange"*, which may or may not be a good thing. One respondent wrote in detail on their thoughts of the virtues of this type of art-science collaboration:

> *"Taking raw data out of context and using it as a key creative element in the creation of art is a way of providing a fresh look at a scientific inquiry. Art can be a mirror whose reflection can reset context and provide the listener with a different perspective than might otherwise be encountered. The result of this competition has been a number of submissions that stimulate a wider audience to think about how science is more than just the collection of raw data, and that understanding can come from looking at results from a new vantage."*

We note that despite the somewhat limited evaluation data, it does not appear that the impacts from events which exhibited the short films (with their prologue and epilogue text concerning the science) are significantly different from those of the anthology film (which contained substantial additional messaging through the bridging film). The overall results highlight that there were many unforeseen impacts upon attendees outside of simply raising awareness of the research area to atypical audiences.

## 6   Conclusions

The SSFX (Space Sound Effects) Short Film Festival was an art-science collaboration project aimed at infiltrating space science into culture through the medium of film. In particular it invited the usage of sonified satellite data of plasma waves in Earth's magnetosphere, a key component within space weather, as key creative elements.

The first audience the project aimed to engage were independent filmmakers through challenging them to use these space sounds to create short films. Through partnership with film industry experts and organisations, an international film festival was run adopting many of the standard practises within the sector to lend authenticity and legitimacy to the project. Formative evaluation of people who registered interest with the project during the submission revealed that we successfully hooked the filmmaking community, though most who engaged also had a general interest in science. Seven very different films were selected for screening. Feedback from these filmmakers highlighted that they relished the creative freedom afforded to them in interpreting the sounds and their usage within their works, hence very open criteria are not only enticing to filmmakers but also enable a broad range of art works to be produced. Another important aspect to the project was in supporting the filmmakers and championing their films after the initial festival, which had the mutual benefit of raising the profile of the filmmakers whilst also sharing the underlying science more widely.

The second audience was film programmers and exhibitors in trying to infiltrate the produced short films into existing events. While an anthology film packaging all the shorts together through a science-based narrative was produced, we struggled to get this shown and found much greater success with the individual short films. Filmmakers were best placed to submit their own works to film festivals following the standard method, with monetary support from the scientists, as they have a better idea of which festivals would be most appropriate. However, scientists were still able to play a role in representing the full suite of shorts for consideration at other sorts of film events. Both of these approaches led to SSFX infiltrating more art events than science ones, as desired, though a substantial number of art-science events also occurred.

The project ultimately also aimed to raise awareness of the science to atypical audiences through the use of the films. While audience evaluation proved challenging due to SSFX films typically sitting within larger events organised by others, some evaluation was able to be done at mostly bespoke art-science events. This highlighted that attendees were much less aware of the topic of space weather than the general public and were much more likely to have attended due to an existing interest in film or specifically the concept of SSFX rather than having an existing science interest. This placed the project as comparable to some of the most successful art-science events across different art forms at reaching new audiences. Many different, and often unanticipated, impacts were had on attendees beyond simply learning about the science, which demonstrates the versatility of film as a form of art at provoking varied responses in audiences.

We therefore advocate that adopting a film festival model can result in creative art-science that fits within the many film-based cultural events around the world. This enables the power of cinema to be leveraged on audiences that don't normally engage with science, thus providing one potential means of breaking beyond the scientific "echo chamber" in perveying the importance and relevance of scientific research.

**Appendix A: Statistical techniques**

Several statistical methods are used throughout this paper which are detailed here.

All uncertainties quoted or displayed, e.g. through errorbars, represent standard (i.e. 68%) intervals. For proportions/probabilities these are determined through the Clopper and Pearson (1934) method, a conservative estimate based on the exact expression

for the binomial distribution, and therefore represent the expected variance due to counting statistics only and not any other potential sources.

Several statistical hypothesis tests are used with effect sizes and two-tailed $p$-values being quoted. Throughout the desired significance level $\alpha$ is set as 0.05, though in the case of multiple comparisons we use the Bonferonni correction where the significance level per comparison is $\alpha_{Bonf} = \alpha/N$ for $N$ total possible comparisons. Two-tailed binomial tests are used to compare proportions of both independent and correlated (i.e. within the same) samples.

Finally, the agreement between judges scores is quantified using the alpha coefficient of Krippendorff (1970, 2018), which is computed as unity minus the ratio of the observed disagreement to that expected by chance, i.e.

$$\alpha = 1 - \frac{\frac{1}{n} \sum_c \sum_k o_{ck} \delta_{ck}^2}{\frac{1}{n(n-1)} \sum_c \sum_k n_c n_k \delta_{ck}^2}$$

where $o_{ck}$ are the observed frequencies in a coincidence matrix, $n_c$ are the column totals in this matrix, $n$ is sum of the entire matrix, and $\delta_{ck}$ is a metric function for which we use the one applicable to ordinal data. The intepretation of this coefficient is that a value of 1 indicates perfect agreement between judges, 0 would result from randomly drawn scores, and a negative value is possible when disagreements are systematic and exceed what can be expected by chance.

## Appendix B: Qualitative data

Here we tabulate the various qualitative data captured from audiences at events, where each row contains responses from a single unique participant. The qualitative data was coded and analysed by the author using thematic analysis (Braun and Clarke, 2006), however, no a priori codes were generated instead allowing these to naturally emerge from the data via a grounded theory approach (Silverman, 2010; Robson, 2011). The final themes determined by this method and their association to the raw qualitative data are also listed in the following tables.

Firstly, the responses from researchers and public engagement professionals to open questions through an interactive online survey during an art-science session at the 2019 Interact symposium (Archer et al., 2019) were as follows.

| Why are you interested in art-science collaborations? | Who do you think are the target audience in art-science? | Why do you think art-science is important? |
| --- | --- | --- |
| I want to build creativity into my role | A wide range of audiences, beyond academia | Both pursuits are creative - potential to communicate in new ways |
| [Blank] | Anyone interested in art. | Different ways of communicating science |
| To communicate abstract concepts | Everyone | Increase fascination and curiosity in BOTH subjects |
| Worked well in the past. Would like to find out more! | People who do not normally engage with science | [Blank] |
| I enjoy both and don't see why I should have to choose between them. | People who think they like art but not science | Science should be an embedded part of culture |
| It's important to combine them together | Everyone should be as science and art is everywhere | It's everywhere whether you realise it or not |

| | | |
|---|---|---|
| I'm a scientist and my husband is an artist. We collaborate and I'm interested to see how others combine the subjects. | Everyone | Communication tool |
| I work with communities that do not have English as their first language and art crosses language barriers | Anyone and everyone | Reaches new audiences and can illustrate the science in new exciting ways |
| Provides new creative accessible ways to access science and vice versa | Depends on purpose of engagement activity | Need to enable conversations and accessibility and tap into interests / challenges |
| It is fun and very creative! | Anyone! Crosses all spectrums | To get a science message across in a unique thought provoking way and challenge artists to enage in new areas |
| To describe science in interesting new ways | Everyone | Because it can explain quite abstract concepts in a unique way |
| Engage with new audience | Everyone | Art and science are both creative both can learn from each other. |

The following table displays the responses from audiences captured on a graffiti wall concerning what attracted them to SSFX art-science events.

| Event | What attracted you to this event? | Themes | | |
|---|---|---|---|---|
| | | Science | Film | Project |
| ab | Collab-lab [factual science filmmakers] tweet + interest in subject | ✓ | | |
| ab | Interesting concept | | | ✓ |
| ab | Email on MIST [Magnetosphere Ionosphere Solar Terrestrial community] | ✓ | | |
| ac | Short films with an awesome original soundtrack | | ✓ | ✓ |
| ac | Interested in the brief to create films from such an unusual sound source | | ✓ | ✓ |
| ac | Space + sound art-science crossover | | | ✓ |
| ac | Love science and sound design | ✓ | | ✓ |
| ac | I like weird films | | ✓ | |
| ac | I wanted to see what filmmakers could do with space sounds | | ✓ | ✓ |

A similar graffiti wall also asked audiences what they felt that they'd gained from attending these events.

| Event | What if anything have you gained or taken away from this event? | Themes | | |
|---|---|---|---|---|
| | | Learn science | Sound perception | Art/Film |
| ab | Ideas on how to explain space weather | | | |
| ab | National anthem as cultural boundary | | | |
| ab | Noise | | ✓ | |
| ab | Space weather | ✓ | | |
| ab | Use sound to promote my research and get $$ | | ✓ | |
| ab | Chirp | | ✓ | |
| ab | People hear differently | | ✓ | |

| ab | Please see "no" ball container | ✓ | | |
| ab | Ideas of being human | | | |
| ab | Grown an interest in film-making | | | ✓ |
| ab | Perceptions of space sounds | | ✓ | |
| ab | .- .-. - (art) is a mirror | | | ✓ |
| ac | Amazing space sounds I want to learn more about it | ✓ | | |
| ac | I want to learn more about the science behind it | ✓ | | |
| ac | I had never heard of SSFX before but now want to hear more | ✓ | | |
| ac | The theme of the narrative linking segments put me in mind of the Hungarian film 'Adas' (Transmission) where all electronics mystreriously die | | | ✓ |

Finally, the results from an online survey three weeks following various SSFX events were as follows.

| Event | What was the theme of the event? | Why is this topic important / studied? | Themes | | Is there anything else you gained or took away from the event? | Any comments you'd like to feed back? | Themes | | | Science |
| --- | --- | --- | --- | --- | --- | --- | --- | --- | --- | --- |
| | | | Space weather | Curiosity | | | Enjoyed / Inspired | Creativity / Diversity | Artists & Scientists | |
| ab | Space Sounds | Space is bigger than anything human and it gives you a very different perspective on the human world and the issues that face us. | | ✓ | The importance of space and continually exploring the unknown. | Taking raw data out of context and using it as a key creative element in the creation of art is a way of providing a fresh look at a scientific inquiry. Art can be a mirror whose reflection can reset context and provide the listener with a different perspective than might otherwise be encountered. The result of this competition has been a number of submissions that stimulate a wider audience to think about how science is more than just the collection of raw data, and that understanding can come from looking at results from a new vantage. | | ✓ | ✓ | ✓ |
| ab | Space sounds | It helps us to understand the universe, physics, and gives us a clearer idea of the world around us | | ✓ | I enjoyed the creativity and diversity shown in the films, and how each filmmaker found the humanity in sounds from space | Very well put together, good programming, enjoyed the presentation at the start. | ✓ | ✓ | | ✓ |
| ab | Sound in space | It's interesting because the films allow people access to science/physics in an accessible way and interpret something that happens quite far away onto a human level. | | ✓ | I will definitely look at the Rich Mix website more for future events | The event wasn't sign posted. Made it a bit tricky to know when we're in the correct place | | | | |
| ab | Science communication using sound recordings from space | Facinating but not important... | | ✓ | Genuine and relevant science research and knowledge is vital and underused in the film industry. | This festival needs to grow and expand! It is important! | | | | ✓ |
| aa | Measuring background magnetic variation around earth | For satellites to operate properly and to improve our understanding of the universe | ✓ | | Really enjoyed the creativity of the films! | Very slickly run, might have been nice to intersperse the talk with the videos rather than all in one. | | ✓ | | |
| aa | Unsure | Unsure | | | N/A | Enjoyable evening, fascinating talk | ✓ | | | ✓ |
| aa | Sounds from space | Gives us ideas about how the universe works | | ✓ | Really enjoyed the enthusiasm of the speaker and the topic of the films mixed with science | Brilliant. | ✓ | | ✓ | ✓ |
| ac | Sounds from Space | Because Space is cool | | ✓ | It was very strange | [Blank] | | | | |
| ac | Life | It was very interesting. | | ✓ | Inspiration. | [Blank] | ✓ | | | |
| h | Space, space storms | To see, gauge potential impact/predict/prevent | ✓ | | An awareness that this existed | [Blank] | | | | ✓ |
| h | Fluctuations in the magnetosphere | To protect satellites and power grids from electromagnetic interference - for example a repeat carrington event | ✓ | ✓ | It was interesting to meet some of the people involved in both the science and filmmaking | We had fun! | ✓ | | ✓ | |
| f | Sound from space used in films | Physics and Films is fascinating | | ✓ | I really enjoyed listening to Dr Martin Archer learned a lot | I hope another one can be organised | ✓ | | ✓ | ✓ |

*Data availability.* Data supporting the findings of this study are contained within the article or derived from listed public domain resources.

*Author contributions.* MOA conceived the project and its evaluation, performed the analysis, and wrote the paper.

*Competing interests.* The author declares that they have no conflict of interest.

*Acknowledgements.* We thank the SSFX Short Film Festival judges Laura Adams, David Berman, Ed Prosser, and Jake Roper; the selected filmmakers Adam Azmy, Victor Galvão, Nidhi Gupta, Aaron Howell, Ali Jennings, Simon Rattigan, Jesseca Ynez Simmons, and James Uren; and all the film industry experts and exhibitors that helped us share this work with audiences. This project was supported by a QMUL
Centre for Public Engagement Large Award, EGU Public Engagement Grant, and STFC Public Engagement Spark Award ST/R001456/1.

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
