# Peer review of "SSFX (Space Sound Effects) Short Film Festival: Using the film festival model to inspire creative art-science and reach new audiences"

_Geoscience Communication, 2020_

## Referee Comment (RC1) · Anonymous Referee #1 · 25 Mar 2020

This paper describes the process of creation of a short film festival inspired by satellite recordings of perturbations of the magnetic field, converted into audio datasets. The paper is interesting to the broad field audience of Geoscience Communication. It is a revealing journey behind the scenes of producing a film festival. The project wisely followed the standard processes of international film festivals, which was important to call independent filmmakers' attention. The project target audiences were independent filmmakers, film programmers and exhibitors, and attendees of film festivals. The author provided some evidence that shows the project was a suitable method to infiltrate space science into culture. Although of good quality, the manuscript could be improved following small suggestions described below.

[Figure]

Suggestions and comments: 1) The introduction section could be improved. The topic introduction on the Abstract is better, in the way it presents the topic. Also, the objectives are not clearly set at the end of the introduction. On the contrary, in the conclusions section, the objectives and audiences became clear; I suggest that that phrasing could be followed at the beginning of the manuscript. 2) The numbers of film competition participants and film exhibitions are on their selves proofs that independent filmmakers, film programmers and exhibitors became, at least, aware of space weather. However, having assisted all films and the anthology, which was an enjoyable part of this paper revision, a question came across. Films are different in more than one way. There is not enough evidence to understand if the impacts of "Saturation" are similar to "Noise". In Table 1 it is not clear if audiences of shorts assisted all films. The anthology adds a message, that films independently exhibited (at least some) do not. This ambiguity can easily be solved adding information to Table 1. 3) I'm not sure the alpha coefficient of Krippendorff (2018) is a straightforward concept for Geoscience Communication audience. The editor can disregard this note if consider otherwise. 4) The gathering of data about impact on festival attendees could have gone further. On page 16, lines 384-385, it is not clear how the "ball in bin questions upon arrival assessing prior knowledge" was actually made. What were the questions? How was the content analysis of the grafitti wall made? The quotes on page 19 seem cherry picked, they do not configure a systematic qualitative assessment of impact. From the science communication research point of view, these methodologies are somewhat fragile. This may be also related to way the paper is structured. There is no traditional narrative introduction - methods - results, which is totally understandable given the type of work, but turn some analysis more difficult to follow. All this information can be added as Appendix. 5) Lines 320-338: the way quotes are presented, not formatted in a different way, hinder the reading. It is not obvious if different goers are being quoted or it is the same person. 6) There is an excess of grey literature in the reference section. Of course, this is an innovative work, which means there is not a solid literature body to build upon. Nevertheless, it is not the first publication about art inspired by science

envisioning science communication; an integrative discussion of this work in light of others would greatly improve the already interesting manuscript and place it within science & art and science communication state-of-the-art. 7) There are some typos that the author can check in a revised version. Not exhaustively: line 81 (amongst), line 74 (? missing), line 92 (specific), line 366 (infiltrating), Table 1 (* next to some numbers not led to a footnote).
* * *

---

## Author Comment (AC1) · 30 Mar 2020

This paper describes the process of creation of a short film festival inspired by satellite recordings of perturbations of the magnetic field, converted into audio datasets. The paper is interesting to the broad field audience of Geoscience Communication. It is a revealing journey behind the scenes of producing a film festival. The project wisely followed the standard processes of international film festivals, which was important to call independent filmmakers' attention. The project target audiences were independent filmmakers, film programmers and exhibitors, and attendees of film festivals. The author provided some evidence

that shows the project was a suitable method to infiltrate space science into culture. Although of good quality, the manuscript could be improved following small suggestions described below.

We thank the reviewer for their careful assessment of the manuscript and address their suggestions for improvement below.

1) The introduction section could be improved. The topic introduction on the Abstract is better, in the way it presents the topic. Also, the objectives are not clearly set at the end of the introduction. On the contrary, in the conclusions section, the objectives and audiences became clear; I suggest that that phrasing could be followed at the beginning of the manuscript.

We agree with the reviewer that the introduction could better frame the following sections, in a similar way to how that is done in the abstract and conclusions. We have added the following paragraph at the end to address this:

This paper concerns a film festival project called SSFX (Space Sound Effects), devised and run by the author, which aimed to integrate space science research into culture. The scientific basis for the project was the ultralow frequency (ULF) analogues of sound present within near-Earth space (Keiling et al., 2016, and references therein) which had been converted into audible sound (Archer et al., 2018). The motivations for choosing to use these sounds for the creation of art, and in particular through film, are discussed in section 2. The SSFX project had two phases, both with different target audiences and aims. Phase one targeted filmmakers, aiming to engage the independent filmmaking community with the sounds present in the near-Earth space environment and enable the creation of creative short films inspired by and incorporating these sounds. This was tackled by running an international short film competition (adopting standard film festival practises through partnering with film industry professionals) which

challenged filmmakers to use the sounds as key creative elements. Section 3 concerns this phase of the project and the subsequent collaborative relationships that formed between scientists and filmmakers through the project. It was through these relationships that phase two of SSFX was possible, which aimed to exhibit these films to wide and diverse audiences, exposing them to this area of space science research with the aim of positively impacting upon these non-traditional audiences. This phase therefore had two target groups, film exhibitors/programmers and independent film-goers. Section 4 discusses how film exhibitors and programmers were engaged to integrate the films into their events and venues, whereas section 5 concerns evaluating the backgrounds of the audiences that attended these events and what impacts resulted from them.

2) The numbers of film competition participants and film exhibitions are on their selves proofs that independent filmmakers, film programmers and exhibitors became, at least, aware of space weather. However, having assisted all films and the anthology, which was an enjoyable part of this paper revision, a question came across. Films are different in more than one way. There is not enough evidence to understand if the impacts of "Saturation" are similar to "Noise". In Table 1 it is not clear if audiences of shorts assisted all films. The anthology adds a message, that films independently exhibited (at least some) do not. This ambiguity can easily be solved adding information to Table 1.

This is a good point. We have added to Table 1 which short films were screened. While this reveals that some were more successful than others, few of these differences are statistically significant. We make a comment in the revised manuscript:

Of the individual shorts 'Astroturf' was the most successful, though the only statistically significant differences ( $\alpha_{Bonf} = 0.0024$ ) in the number of events/initiatives by film were between 'Astroturf' and both 'Names and

Numbers'  $(p = 5.0 \times 10^{-4})$  and 'Saturation'  $(p = 1.9 \times 10^{-4})$ . We note that neither of these latter two films' festival submission fees were funded by the project and in the case of 'Saturation' a number of exhibitors expressed that they could not screen it at their family-friendly events due to the potentially upsetting medical imagery (edited clips from 'Noise' removing the strong language and drug usage were however able to be used).

We are not able to state whether the different short films had different impacts upon audiences, which we explain as follows:

Given that these events where evaluation was possible tended to show all the shorts (either individually or via the anthology) we are unable to comment on whether certain SSFX films were more impactful upon attendees than others.

Finally, we note that there did not appear to be a difference in impact from the data collected between exhibiting the individual shorts or the anthology film, which we now note:

We note that despite the somewhat limited evaluation data, it does not appear that the impacts from events which exhibited the short films (with their prologue and epilogue text concerning the science) are significantly different from those of the anthology film (which contained substantial additional messaging through the bridging film).

3) I'm not sure the alpha coefficient of Krippendorff (2018) is a straightforward concept for Geoscience Communication audience. The editor can disregard this note if consider otherwise.

The author raises a good point about this measure of agreement not being familiar to the journal's audience. We now clarify in the text how to interpret the value of the coefficient stating that a value of 1 would indicate perfect agreement whereas 0 would result from randomly drawn scores. We also elaborate on this measure in the appendices as follows:

Finally, the agreement between judges scores is quantified using the alpha coefficient of Krippendorff (1970, 2018), which is computed as unity minus the ratio of the observed disagreement to that expected by chance, i.e.

$$\alpha = 1 - \frac{\frac{1}{n} \sum_{c} \sum_{k} o_{ck} \delta_{ck}^2}{\frac{1}{n(n-1)} \sum_{c} \sum_{k} n_c n_k \delta_{ck}^2}$$

where  $o_{ck}$  are the observed frequencies in a coincidence matrix,  $n_c$  are the column totals in this matrix, n is sum of the entire matrix, and  $\delta_{ck}$  is a metric function for which we use the one applicable to ordinal data. The intepretation of this coefficient is that a value of 1 indicates perfect agreement between judges, 0 would result from randomly drawn scores, and a negative value is possible when disagreements are systematic and exceed what can be expected by chance.

**4) The gathering of data about impact on festival attendees could have gone further. On page 16, lines 384-385, it is not clear how the "ball in bin questions upon arrival assessing prior knowledge" was actually made. What were the questions?**

We apologise for the confusion here, lines 384–386 were simply intended to summarise the various approaches used to evaluate the events with the following paragraphs providing more detail. In the case of the ball and bin method, only one question was asked though at multiple events. This pertained to audiences' prior knowledge, discussed on lines 390–392. We now make this more explicit stating:

We therefore asked audiences upon arrival at events whether they had heard of space weather before, via a ball and bin method where attendees were instructed to put a ball in either the 'yes' or 'no' bin.

How was the content analysis of the grafitti wall made? The quotes on page 19 seem cherry picked, they do not configure a systematic qualitative assessment of impact. From the science communication research point of view, these methodologies are somewhat fragile. This may be also related to way the paper is structured. There is no traditional narrative introduction - methods - results, which is totally understandable given the type of work, but turn some analysis more difficult to follow. All this information can be added as Appendix.

A systematic qualitative coding of all gathered data was peformed using grounded theory analysis, which we now make more explicit throughout in the text. Since the quantity of qualitative data collected was not exhaustive, we opt to now tabulate this data in an appendix along with a brief discussion of the analysis performed.

Here we tabulate the various qualitative data captured from audiences at events, where each row contains responses from a single unique participant. The qualitative data was coded and analysed by the author using thematic analysis (Braun and Clarke, 2006), however, no a priori codes were generated instead allowing these to naturally emerge from the data via a grounded theory approach (Silverman, 2010; Robson, 2011). The final themes determined by this method and their association to the raw qualitative data are also listed in the following tables.

To address in the text that the quotes included are simply included to illustrate the emergent themes we note

where the quotes displayed serve as representative illustrative examples from different respondents

which the reader can check by examining the appendix.

5) Lines 320-338: the way quotes are presented, not formatted in a different way, hinder the reading. It is not obvious if different goers are being quoted or it is the same person.

We have adapted how extensive quotes are formatted using quote blocks to make them clearer and, where possible, including an anonymous identifier of the individual.

6) There is an excess of grey literature in the reference section. Of course, this is an innovative work, which means there is not a solid literature body to build upon. Nevertheless, it is not the first publication about art inspired by science envisioning science communication; an integrative discussion of this work in light of others would greatly improve the already interesting manuscript and place it within science & art and science communication state-of-the-art.

We performed an extensive literature search, seeking relevant discussions and analysis to the aims and results presented in the manuscript, e.g. the backgrounds of attendees at art-science events, finding that the most pertinent work came from grey literature rather than standard papers. However, we take the reviewers point and have added a short discussion to the introduction illustrating the breadth of science-inspired art from published literature.

There have been numerous published examples of science-inspired artworks (Type V), where science acts as a resource for creative art (Kim, 2011). Voss-Andreae (2011) presents sculptures inspired by quantum physics that he argues can indicate aspects of reality that science cannot. The Tumamoc Hill Arts Initiative was a collection of site-based art and writing inspired by the Sonoran Desert and the underlying science of the region Mirocha et al. (2015). Similarly, Orfescu (2012) describes artistic interpretations of scientific images, in this instance nanostructures, where artists convert them into pieces of art. Hoare (2013) posits that even classic works of literature, such as 'Moby Dick', have strong scientific influences since art and science were not strictly demarcated at the time. It is therefore clear, even from these few examples, that activities attempting to integrate science into culture are incredibly varied and have been undertaken for a long time.

**7) There are some typos that the author can check in a revised version. Not exhaustively: line 81 (amongst), line 74 (? missing), line 92 (specific), line 366 (infiltrating), Table 1 (\* next to some numbers not led to a footnote).**

We have corrected typographical errors.

**References**

- Archer, M. O., Hartinger, M. D., Redmon, R., Angelopoulos, V., Walsh, B. M., and Eltham Hill School Year 12 Physics students: First results from sonification and exploratory citizen science of magnetospheric ULF waves: Long-lasting decreasing-frequency poloidal field line resonances following geomagnetic storms, Space Weather, 16, 1753–1769, https://doi.org/ 10.1029/2018SW001988, 2018.
- Braun, V. and Clarke, V.: Using thematic analysis in psychology, Qualitative Research in Psychology, 3, 2006.
- Hoare, P.: Cetology: How science inspired Moby-Dick, Nature, 493, 160–161, https://doi.org/ doi.org/10.1038/493160a, 2013.
- Keiling, A., Lee, D.-H., and Nakariakov, V., eds.: Low-Frequency Waves in Space Plasmas, Geophysical Monograph Series, American Geophysical Union, https://doi.org/10.1002/ 9781119055006, 2016.
- Kim, S.: Art, Science and the Public: Focusing on Science as a Resource for Creative Art, in: EKC 2010, edited by Han, M. W. and Lee, J., vol. 138 of *Springer Proceedings in Physics*, Springer, Berlin, Heidelberg, https://doi.org/10.1007/978-3-642-17913-6\_6, 2011.

Krippendorff, K.: Estimating the reliability, systematic error, and random error of interval data, Educational and Psychological Measurement, pp. 61–70, https://doi.org/10.1177/001316447003000105, 1970.

Krippendorff, K.: Content Analysis An Introduction to its Methodology, SAGE, 4 edn., 2018.

- Mirocha, P., Magrane, E., Terkanian, B., Soria, M., Coleman, D. L., Milstead, M., and Koopman, K.: The Tumamoc Hill Arts Initiative: A Portfolio of Site-Based Art and Writing Inspired by a History of Sonoran Desert Science, Journal of the Southwest, 57, https://www.jstor.org/ stable/26310169, 2015.
- Orfescu, C.: Biologically-Inspired Computing for the Arts: Scientific Data through Graphics, chap. NanoArt: Nanotechnology and Art, IGI Global, https://doi.org/10.4018/ 978-1-4666-0942-6.ch008, 2012.

Robson, C.: Real World Research, John Wiley and Sons Ltd., 2011.

Silverman, D.: Doing Qualitative Research: A Practical Handbook, Sage, 2010.

Voss-Andreae, J.: Quantum Sculpture: Art Inspired by the Deeper Nature of Reality, Leonardo, 44, 14–20, https://doi.org/10.1162/LEON\_a\_00088, 2011.

---

## Referee Comment (RC2) · Antonio Menghini (Referee) · 18 May 2020

The paper is well written and it's nicely balanced among the presentation of the project, its aim and its outcome: the choice to describe each video provides a more "artistic" flavor to the paper, without neglecting the scientific side, as such as the use of statistics that could be too heavy for standard reader (it is a good idea also to detail any statistics issue in an appendix). Ad associate editor I had already suggested some editing in a preliminary phase, that have been positively accepted by the author, hence I think the paper can be considered ready for publication now.

---

## Author Comment (AC2) · 18 May 2020

We thank the reviewer for their considered comments on and assessment of the paper.